# A BaSiC tool for background and shading correction of optical microscopy images

Tingying Peng[1,2,3], Kurt Thorn[4], Timm Schroeder[5], Lichao Wang[1,2], Fabian J. Theis[2,3], Carsten Marr[2] & Nassir Navab[1,6]

Quantitative analysis of bioimaging data is often skewed by both shading in space and background variation in time. We introduce BaSiC, an image correction method based on low-rank and sparse decomposition which solves both issues. In comparison to existing shading correction tools, BaSiC achieves high-accuracy with significantly fewer input images, works for diverse imaging conditions and is robust against artefacts. Moreover, it can correct temporal drift in time-lapse microscopy data and thus improve continuous single-cell quantification. BaSiC requires no manual parameter setting and is available as a Fiji/ImageJ plugin.

[1] Department of Computer Science, Chair of Computer Aided Medical Procedure, Technische Universität München, Boltzmannstr. 3, Garching 85748, Germany. [2] Institute of Computational Biology, Helmholtz Zentrum München—German Research Center for Environmental Health, Institute of Computational Biology, Ingolstädter Landstraße 1, Neuherberg 85764, Germany. [3] Center for Mathematics, Chair of Mathematical Modeling of Biological Systems, Technische Universität München, Boltzmannstr. 3, Garching 85748, Germany. [4] Department of Biochemistry and Biophysics, University of California, San Francisco, 600 16th Street, San Francisco, California 94158, USA. [5] Department of Biosystems Science and Engineering (D-BSSE), ETH Zurich, Basel 4058, Switzerland. [6] Department of Computer Science, Chair of Computer Aided Medical Procedure, Johns Hopkins University, 3400 North Charles Street, Baltimore, Maryland 21218, USA. Correspondence and requests for materials should be addressed to C.M. (email: carsten.marr@helmholtz-muenchen.de) or to N.N. (email: nassir.navab@tum.de).

O ptical imaging is an indispensable tool in biomedical research. All modern optical imaging (whether whole slide imaging, high-content screening or high-throughput time-lapse microscopy) relies on image processing and quantification methods to analyse and interpret the acquired data. However, optical microscopy data, and especially fluorescence imaging, is often severely affected by shading or vignetting[1], typically reflected as an attenuation of the brightness intensity from the centre of the optical axis to the edges. This not only degrades the visual quality of an image (for example, by causing discontinuities in whole slide images (WSIs)), but more critically compromises the downstream analysis of, for example, tissue composition or single-cell properties. Besides spatial shading effects, time-lapse movies often exhibit a temporal baseline drift due to background bleaching, which further skews the quantification of the dynamic behaviour of cellular and molecular properties[2].

The physical process of image formation can be approximated as a linear function[3] that relates a measured image, $I^{meas}(x)$ at location $x$, to its uncorrupted true correspondence, $I^{true}(x)$, as

$$I^{meas}(x) = I^{true}(x) \times S(x) + D(x) \qquad (1)$$

where the multiplicative term $S(x)$ represents the change in effective illumination across an image (known as flat-field); the additive term $D(x)$, known as dark-field, is dominated by camera offset and thermal noise, which are present even if no light is incident on the sensor.

Existing shading correction methods can be generally divided into two groups: 'prospective' approaches that determine $S(x)$ and $D(x)$ from extra reference images[4] (Supplementary Note 1) and 'retrospective' approaches, which rely on the actual image data itself and hence avoid collection of extra reference images (Supplementary Note 2). A number of multi-image based approaches have been recently published, for example, Smith et al.[5] (Fiji Plugin 'CIDRE'), Coster et al.[6] and Singh et al.[7] (the default module in CellProfiler). These approaches take advantage of shared $S(x)$ among an image sequence and are usually more reliable than single-image based corrections. Yet they require large numbers of images to reach a stable performance and a manual fine-tuning of internal parameters, and their robustness to common bioimage artefacts (such as dust and fluorescence dye particles) has not been tested. Moreover, none of the existing methods is able to model and correct temporal drift (e.g. caused by photobleaching) for time-lapse movies.

We propose BaSiC, a retrospective method for background and shading correction of image sequences, based on a sparse and low-rank decomposition. In comparison to existing shading correction tools, BaSiC requires fewer input images, works for diverse imaging conditions and is robust against typical image artefacts. Moreover, it can correct temporal drift for time-lapse microscopy data, and hence improve single-cell quantification. BaSiC is available as an easy-to-use Fiji/ImageJ plugin as usually requires no manual parameter tuning.

## Results

### BaSiC workflow

Inspired by Smith et al.[5], we build our method on the shading model (equation (1)), which accounts for the effect of both $S(x)$ and $D(x)$. Such a full model is superior as compared to a partial model that considers $S(x)$ only[5]. As shown in the schematic plot of Fig. 1, BaSiC first constructs a measurement matrix $\mathbf{I}$ (step I), which is then decomposed into a low-rank matrix $\mathbf{I^B}$ and a sparse residual matrix $\mathbf{I^R}$ (step II). The low-rank matrix has a maximum rank of two as each column is the sum of a scaled version of $S(x)$ (with a scaling factor $B_i$) and $D(x)$, which are all initialized with zeros (step III) and optimized by promoting the sparsity of the residual matrix with a

reweighted L1-norm (step IV). In addition, smooth constraints are imposed on both $S(x)$ and $D(x)$ by regulating their sparsity in Fourier domain (Supplementary Fig. 1). The optimization is solved in an iterative fashion using the linearized augmented Lagrangian method[8], which is widely used in sparse matrix decomposition like Robust PCA (ref. 9) and RASL (ref. 10) (for a detailed description of the mathematical derivation and matrix updating see Methods and Supplementary Note 3). An automatic parameter setting strategy determines the smooth regularization parameters for $S(x)$ and $D(x)$, adaptive to different image contents, so that tedious manual parameter tuning is avoided (Supplementary Note 4). We provide a statistical interpretation of BaSiC in Supplementary Note 5. With the estimation of $S(x)$ and $D(x)$, we can correct the intensity profile of each image tile of a WSI by reversing the image formation process, equation (1), leading to a homogenous appearance and correct stitching (Fig. 1d versus a).

### BaSiC requires few images for shading correction

We first evaluate BaSiC using synthetic images, where the ground-truth of the shading-free image $I^{true}$, the flat-field $S(x)$ and the dark-field $D(x)$ are known (Supplementary Note 6). We first compare BaSiC with CIDRE (ref. 5), the only other method that is able to simultaneously estimate $S$ and $D$. BaSiC requires far fewer images to achieve the same accuracy as CIDRE (for example, 10 versus 100 images to reach an estimation score $\Gamma(S^{est}) \leq 0.1$ at three levels of cell density (cell density, Supplementary Fig. 2). We subsequently evaluate BaSiC using a comprehensive microscope image collection provided by Smith et al.[5], which includes 10 real microscopy data sets, one photography data set and one synthetic microscopy data set (refer to Smith et al.[5] for data set details). We assess the correction quality by the correction score, $\Gamma'(I^{corr})$, which is the mean absolute difference of pixel pairs in overlapping image regions for each data set after correction, normalized by the difference of the uncorrected pairs[5]. A $\Gamma'(I^{corr}) < 1$ suggests reduced shading while $\Gamma'(I^{corr}) > 1$ implies increased shading compared to the uncorrected images. Besides CIDRE, we compare BaSiC also with the methods used in Coster[6] and Cellprofiler[7] (refer to Methods for technical details), as well as two prospective methods (Calib-zero and Empty-zero)[5]. We find that BaSiC improves image quality with as few as five images (Fig. 2a). At around 100 images it matches prospective methods on average, although the performance varies for different data sets (Supplementary Fig. 3). Among retrospective methods, BaSiC significantly outperforms existing approaches, when fewer than 500 images are used (Fig. 2a). This is practically relevant since WSI acquisition typically contains 50–200 tiles and high-content screening usually works with 96 and 384 well plates, where only images at the same position of each well share a shading profile[6]—hence the number of images available for one estimation is 96 or 384.

### BaSiC is robust to typical image artefacts

Furthermore, we evaluate the robustness of the four considered retrospective methods with respect to typical image artefacts. One type of common artefacts in fluorescence images is bright particles that strongly fluoresce or scatter light (Fig. 2b). BaSiC is robust to these artefacts as it incorporates them in the sparse residual without affecting the low-rank estimation of $S(x)$ and $D(x)$ (Fig. 2c). By contrast, the estimated flat-fields $S^{est}(x)$ from CIDRE and Cellprofiler are sensitive to outliers. While existing methods suffer from artefacts and inhomogeneities at the image edges, BaSiC correction leads to a homogenous distribution of mean cell intensities in a cell culture WSI over the whole slide (Fig. 2d). Other typical artefacts are stray light and residual excitation light

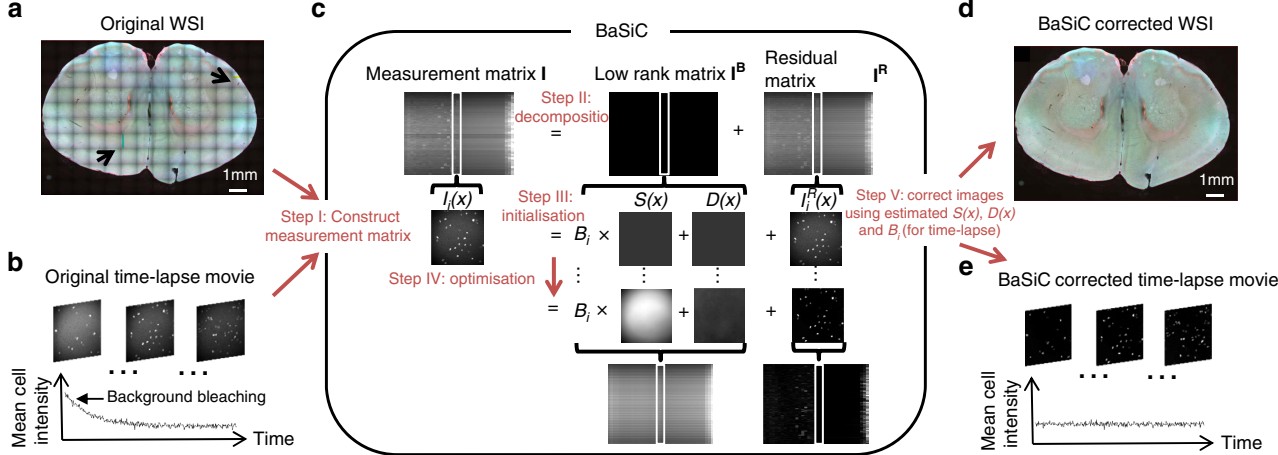

**Figure 1 | BaSiC is an automatic correction method for both static multi-image microscopy data and dynamic time-lapse data.** (**a**) A mosaic whole slide image (WSI) of a mouse brain slice showing intensity discontinuities and resulting stitching errors (indicated by black arrows). (**b**) A time-lapse movie corrupted by both shading in space and photobleaching in time. (**c**) The BaSiC workflow, (see text for detailed explanation). (**d**) The corrected WSI shows improved stitching with no discontinuities. (**e**) BaSiC corrects both spatial shading and background bleaching over time.

due to imperfect filtering, which are difficult to measure experimentally and hence difficult to correct with prospective methods (Supplementary Fig. 4a). BaSiC incorporates such artefacts in the estimation of $D(x)$ and can successfully correct their effect (Supplementary Fig. 4b). A quantitative evaluation of 45 WSIs using the estimation score suggests that BaSiC achieves an accurate estimation of shading in all instances, outperforming existing retrospective methods (Fig. 2e, Supplementary Figs 5–8).

**BaSiC corrects background variation in time-lapse movies.** Finally, we apply BaSiC to improve single-cell quantification of long-term time-lapse microscopy. We decompose the shading-free true image $I_i^{\text{true}}(x)$ of the $i$th frame of a time-lapse microscopy movie into the sum of a spatially-constant baseline signal, $B_i$, and the spatially varying foreground (fluorescence) signal of biological relevance[6]. Hence the full model for a time-lapse movie becomes:

$$I_i^{\text{meas}}(x) = (B_i + F_i(x)) \times S(x) + D(x) \qquad (2)$$

Because of background bleaching and varying experimental conditions, $B_i$ is usually not constant between frames (Fig. 1b). We correct the intensity profile of each frame using the estimated $S(x)$, $D(x)$ and $B_i$ by reversing equation (2), which removes both spatial shading effects and the temporal drift (Fig. 1e versus b).

Continuous monitoring of single-cell differentiation dynamics is an important research tool for stem cell research[11]. Besides improving image contrast at the plate edge, BaSiC is able to remove intensity spikes for bright-field images and photo bleaching of the background medium in the fluorescence channel (Supplementary Fig. 9, Supplementary Movies 1 and 2). We apply BaSiC on 6 day time-lapse movies of hematopoietic stem and progenitor cells that differentiate towards the granulocyte-macrophage (GM) lineage and the megakaryocyte-erythrocyte (MegE) lineage (Fig. 3a, see Hoppe et al.[12] for experimental details). The dynamic expression of the transcription factor PU.1 has been quantified in cells over many generations. The BaSiC-corrected intensity profile illustrates a 2–5-fold increase of PU.1 intensity for GM cells at the onset of the lineage marker CD16/32 (Fig. 3b). In contrast, PU.1 levels stay roughly constant in MegE cells, when the onset of the lineage marker Gata1 is observed (Fig. 3c). Importantly, the uncorrected intensity profiles exhibit no obvious change in PU.1 behaviour for the two lineages. When comparing GM versus MegE branches a significant ($P = 1.2 \times 10^{-3}$, Wilcoxon rank-sum test, Fig. 3d)

fold-change in PU.1 expression is only observable after BaSiC correction.

## Discussion

BaSiC is an efficient tool for image correction and can be applied to high-content images, WSIs and high-throughput time-lapse movies. BaSiC has immediate attraction to researchers who create stitched images, since correcting uneven illumination improves stitching and mosaic image quality. Besides, BaSiC can be also used as a pre-processing step in conjunction with automatic methods such as cell counting or measuring the morphology of cells and thus improving down-stream analysis. The crucial contribution of BaSiC is to improve intensity quantification in both static and time-lapse imaging data. Unlike local contrast equalization methods, which could distort the true intensity variations within an original image or across multiple images, BaSiC is built on solid physical models of optical imaging and hence is able to recover biologically relevant intensities for image quantification. Besides of being accurate, BaSiC is also fast to compute: in our Fiji implementation, it usually processes hundreds of images within minutes on a standard laptop.

From a methodological point of view, there are two key differences between BaSiC and the state-of-the-art shading correction tools that also model flat-field $S(x)$ and dark-field $D(x)$. The first distinctive feature of BaSiC is the reweighted L1-norm error measure, which allows for a quicker convergence when dealing with limited amount of images and, more importantly, results in increased resistance to outliers in data such as noise or debris. Besides $S(x)$ and $D(x)$, BaSiC can also estimate a per-image baseline $B_i$, which accounts for varying background in time-lapse movies. This correction of background bleaching is a unique feature of BaSiC that CIDRE and other existing methods cannot provide.

As any shading correction method, BaSiC has limitations. One key assumption of BaSiC and all other previously mentioned multi-image based retrospective methods is that the foreground of every image to be processed should be uncorrelated with the foreground of every other image. This assumption can be violated for time-lapse movies of static and quasi-static objects, for example, for a single-cell of high-magnification that is always in the centre of the field of view. In such cases, BaSiC would consider the consistently higher image intensities in the centre of

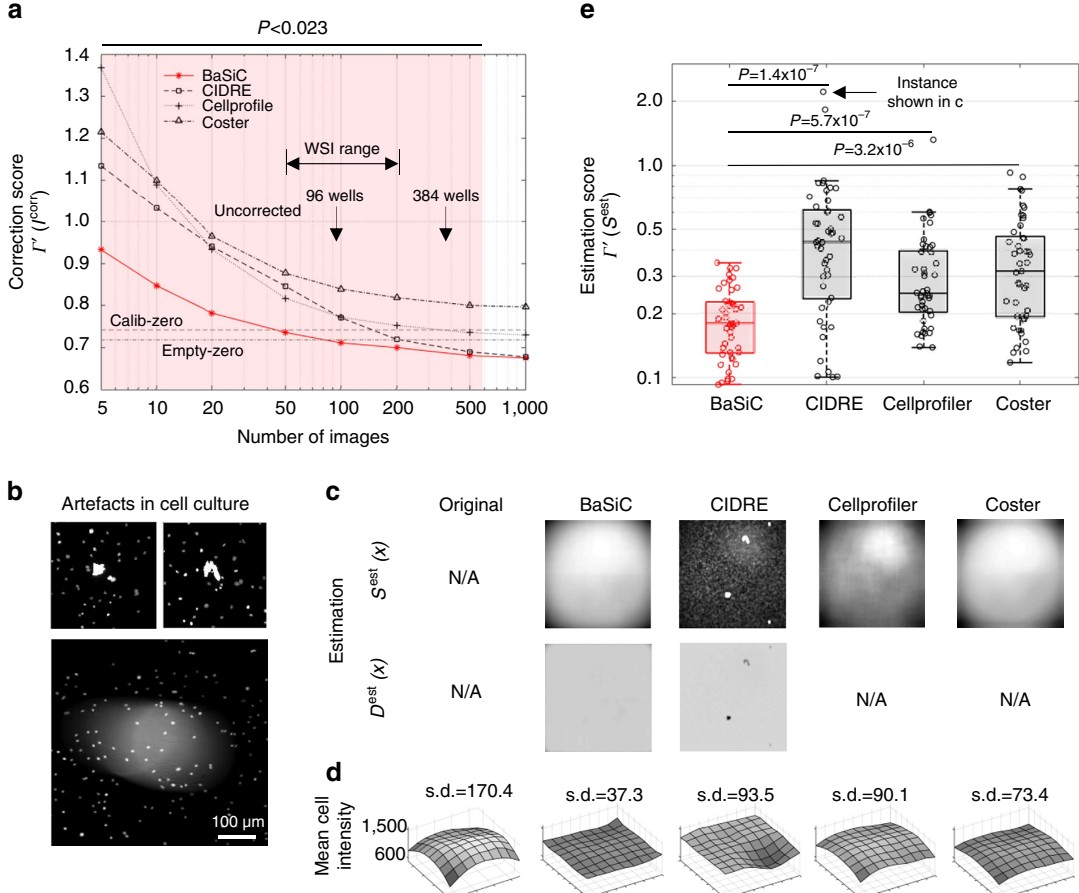

**Figure 2 | BaSiC requires significantly fewer images and is robust to image artefacts. (a)** Method evaluation on benchmark data sets[5]. Correction quality is evaluated by the correction score, $\Gamma'(I^{corr})$, the mean absolute difference between overlapping regions after correction, normalized by the difference before correction. A lower correction score suggests better correction results. Each curve represents the average performance on 12 microscope image collections. For less than 500 images, BaSiC significantly outperforms other retrospective methods ($P < 0.023$, paired Wilcoxon signed rank test with Bonferrnoi correction for multiple testing). With $> 100$ images, BaSiC outperforms two prospective methods (Calib-zero and Empty-zero). **(b)** Typical WSI artefacts: bright particles on the specimen that strongly emit light ('spike'-like artefact, top) or scatter light (bottom). **(c)** BaSiC is more robust to the 'spike'-like artefacts as compared to CIDRE and Cellprofiler. Note that only BaSiC and CIDRE can estimate dark-field, while other methods simply neglect dark-field. **(d)** Surface plots show the averaged cell intensity of different spatial locations before and after shading correction of cell culture WSI. After BaSiC correction, the s.d. of the mean cell intensity over different regions is reduced to around 20% of the uncorrected s.d. **(e)** BaSiC significantly outperforms other methods on 45 WSIs of various types of biological specimen, imaged at different channels and diverse experimental settings, containing noise and artefacts. The estimation score, $\Gamma(S^{est})$, is the mean absolute difference between the estimated $S^{est}(x)$ and the prospectively obtained reference $S(x)$, normalized by a baseline score. Besides achieving the minimum median score, BaSiC, unlike other methods, does not have a single outlier in all cases (refer to Supplementary Figs 5–8 for the individual scores for all 45 WSIs).

the field of view as a local increase in $S(x)$, causing removal of the true fluorescence variability. Nevertheless in practice, BaSiC has some tolerance to such correlations, for example, it performs well in a movie of proliferating and slowly moving embryonic stem cell colonies (as shown in Supplementary Movie 2), in which consecutive frames are correlated. Meanwhile, the regularization parameter $\lambda_s$ (see Methods) can be used to tune the resulting model so that it is more suitable for correlated images. Larger values of $\lambda_s$ lead to a smoother estimation of the low-rank component, thus rejecting small static objects in the estimated $S(x)$. Another useful strategy is to take samples with a large time gap in between to make images less correlated. In any case, we advise users to visually inspect the estimated shading profiles before making a correction in such challenging cases: a smooth $S(x)$ usually indicates a good shading correction, while local inhomogeneities that come from highly corrected foreground objects are a hint of non-optimal correction.

Although BaSiC can compensate background variation, no matter if it is caused by bleaching or by switching microscopy settings, it does not account for variation in the foreground sample fluorescence that may also occur due to photobleaching. In the presented long-term single-cell time-lapse measurements, the dominant corrupting factor is the background variation caused by medium bleaching. Hence subtraction of background bleaching greatly improves the intensity quantification of single cells (as shown in Fig. 3). In fact, existing photobleaching correction methods (such as the Bleaching Correction Plugin in Fiji/ImageJ) are not suitable for correcting foreground cell bleaching in our movies: these methods either assume constant intensity or stable intensity distribution of each frame, which is certainly not the case for transcription factor expression during cell differentiation, where the signal varies depending on the cell type and time. It should also be noted that for fluorescence images, the estimated baseline can converge to the foreground,

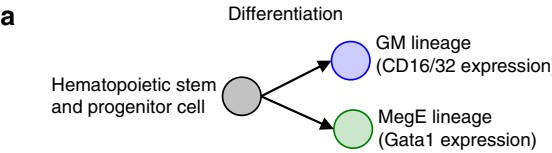

**a**

Differentiation

Hematopoietic stem and progenitor cell

GM lineage (CD16/32 expression)

MegE lineage (Gata1 expression)

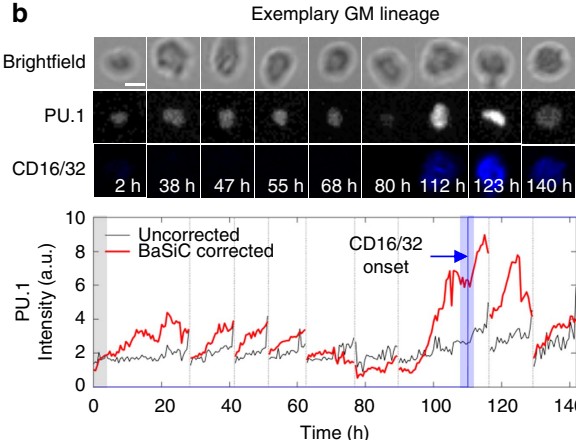

**b** Exemplary GM lineage

Brightfield

PU.1

CD16/32

2 h　38 h　47 h　55 h　68 h　80 h　112 h　123 h　140 h

— Uncorrected
— BaSiC corrected

CD16/32 onset

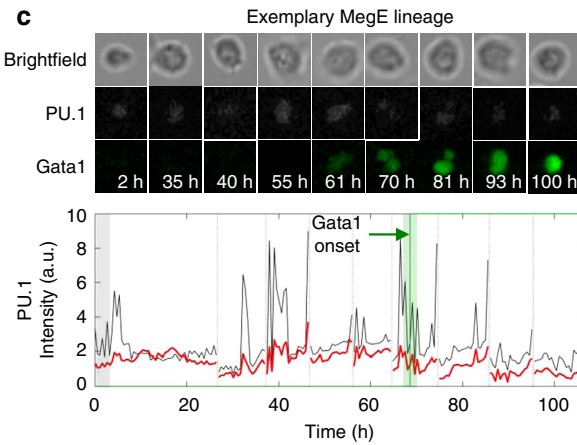

**c** Exemplary MegE lineage

Brightfield

PU.1

Gata1

2 h　35 h　40 h　55 h　61 h　70 h　81 h　93 h　100 h

Gata1 onset

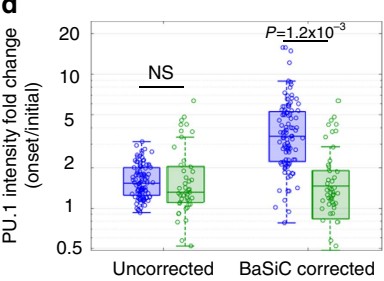

**d**

$P=1.2\times10^{-3}$

NS

**Figure 3 | Single-cell quantification of time-lapse data is significantly improved after BaSiC correction.** (**a**) Hematopoietic stem and progenitor cells can differentiate into two different lineages (GM or MegE). (**b,c**) PU.1 expression along two exemplary cellular branches, one committing to the GM lineage signified by CD16/32 onset (**b**) and the other committing to the MegE lineage signified by Gata1 onset (**c**). The BaSiC-corrected intensity profiles (red) show a PU.1 upregulation in the GM lineage prior to CD16/32 onset, but not in the MegE lineage. The uncorrected profiles (black) suffer from shading, temporal drift and noise. Bright-field, PU.1, CD16/32 and Gata1 image patches of each cell in the branch after BaSiC correction are shown on top (scale bar, 10 μm). (**d**) The BaSiC-corrected PU.1 intensity fold change (right bars) between marker onset and start of the movie (averaged over 6 h) suggests that GM cell branches show a significant ($n=99$, $P=0.0012$, Wilcoxon rank-sum test) PU.1 upregulation in contrast to MegE cell branches ($n=49$). This is not observable in the uncorrected fold changes (left bars).

when the foreground fraction of an image is > 50%. This does not affect the practical usage of BaSiC, when a high-cell density is reached only at the end of a movie. Typically then, the bleaching effect is already weak (bleaching usually decays exponentially), and hence the correction for those frames can be skipped. By contrast, for bright-field images, BaSiC is robust to different levels of cell density in background correction.

With the limitations addressed above in mind, we believe that BaSiC will help to standardize the processing and quantification of bioimage data due to its broad applicability, robust performance, elegant mathematical formulation and easy-to-use interface.

## Methods

**Shading model and optimization.** A measured image sequence, $I^{\mathrm{meas}}=I_1^{\mathrm{meas}}(x)$, ..., $I_n^{\mathrm{meas}}(x)$, can be related to its uncorrupted true correspondence, $I^{\mathrm{true}}=I_1^{\mathrm{true}}(x),...,I_n^{\mathrm{true}}(x)$, with a multiplicative flat-field $S(x)$ and an additive dark-field $D(x)$:

$$I_i^{\mathrm{meas}}(x)=I_i^{\mathrm{true}}(x)\times S(x)+D(x) \qquad (3)$$

The BaSiC correction begins by sorting the image sequence $I^{\mathrm{meas}}$ into $I^{\mathrm{sort}}$ by intensities at each pixel $x$, converting each sorted image $I_i^{\mathrm{sort}}(x)$ into a column vector $\mathbf{I_i^{sort}}$ (from now on, we denote the same parameter in image space with $(x)$ and as vector without $(x)$). Hence, we construct the measurement matrix as

$$\mathbf{I}=\left[\mathbf{I_1^{sort}},...,\mathbf{I_n^{sort}}\right].$$

Each column vector of the measurement matrix $\mathbf{I}$ is decomposed into

$$\mathbf{I_i}=B_i\times\mathbf{S}+\mathbf{D}+\mathbf{R_i}, \qquad (4)$$

where $B_i$ is a location independent scalar and $\mathbf{R_i}$ is the residual. The sum of the first two terms forms a *rank 2* matrix, $\mathbf{I^B}=\mathbf{B}\odot\mathbf{S}\oplus\mathbf{D}$ ($\odot$ and $\oplus$ denote column-wise multiplication and addition, respectively), as all columns share the same $\mathbf{S}$ and $\mathbf{D}$. The residual matrix, $\mathbf{I^R}$, is assumed to be *sparse*, that is, the residual $I_i^R(x)$ generally occupies only a small fraction of image pixels. Assuming sparsity of $\mathbf{I^R}$, we have the following constraint optimization problem:

$$\min_{\mathbf{B,S,D}}\left|\mathbf{I^R}\right|_0 \\ \text{subject to } \mathbf{I}=\mathbf{I^B}+\mathbf{I^R},\mathbf{I^B}=\mathbf{B}\odot\mathbf{S}\oplus\mathbf{D}, \qquad (5)$$

where $|\ |_0$ denotes the L0-norm, that is, the number of non-zero elements in $\mathbf{I^R}$.

A direct optimization of equation (5) is impossible since it has no unique mathematical solution (suppose $\mathbf{D}^\star$ is one solution, then $\mathbf{D}^\star-z\times\mathbf{S},z\in\mathscr{R}$ is another solution). Besides, the minimization of an L0-norm is NP-hard[13]. Hence, we adapt the objective function by imposing regularization on $\mathbf{S}$ and $\mathbf{D}$, and replace the L0-norm by a reweighted L1-norm minimization:

$$\min_{\mathbf{B,S,D}}\left\{\left|\mathbf{W}\circ\mathbf{I^R}\right|_1+\lambda_s\left|\mathcal{F}(S(x))\right|_1+\lambda_d\left|\mathcal{F}(D^R(x))\right|_1+\lambda_d\left|D^R(x)\right|_1\right\} \\ \text{subject to } \mathbf{I}=\mathbf{I^B}+\mathbf{I^R},\mathbf{I^B}=\mathbf{B}\odot\mathbf{S}\oplus\mathbf{D} \\ \mathbf{D}=D^Z+\mathbf{D^R},D^Z\in[0,\min(\mathbf{I})], \qquad (6)$$

where the dark-field $\mathbf{D}$ is decomposed into the sum of its mean $D^Z$ and the residual $\mathbf{D^R}$. $\mathbf{W}$ is the weighting matrix which balances the penalty of large coefficients and small coefficients and hence better approximates the L0-norm[14]. The detailed setting of $\mathbf{W}$, the mathematical derivation of equations (4–6) and the numerical solution of equation (6) are included in Supplementary Note 3. We have developed a strategy to automatically determine the regularization parameters, $\lambda_s$ and $\lambda_d$, adaptive to different image content, so that tedious manual parameter tuning is avoided (Supplementary Note 4). We also provide a statistical interpretation of BaSiC in Supplementary Note 5. With the estimated $S(x)$ and $D(x)$, we can invert the image formation process and obtain corrected image:

$$I^{\mathrm{corr}}(x)=(I^{\mathrm{meas}}(x)-D(x))/S(x). \qquad (7)$$

After sorting intensities, the estimated $B_i$ alongside $S(x)$ and $D(x)$ is no longer the baseline of the original images. Hence, we introduce a two-step strategy to estimate $B_i$: the first step is to compute $S(x)$ and $D(x)$ only, using the matrix with sorted intensities; in the second step, we estimate $B_i$ using the unsorted matrix, $\mathbf{I}=[\mathbf{I_1^{meas}},...,\mathbf{I_n^{meas}}]$ and the estimated $S(x)$ and $D(x)$ from step 1 as model inputs. Hence the optimization problem becomes:

$$\min_{\mathbf{B}}\left|\mathbf{W}\circ\mathbf{I^R}\right|_1 \\ \text{subject to } \mathbf{I}=\mathbf{I^B}+\mathbf{I^R},\mathbf{I^B}=\mathbf{B}\odot\mathbf{S}^\star\oplus\mathbf{D}^\star \qquad (8)$$

where $\mathbf{S}^\star$ and $\mathbf{D}^\star$ are the solutions of equation (6) using sorted images. In comparison to equation (6), solving equation (8) numerically is much faster, due to a reduced degree of freedom. This is practically beneficial as it can reduce the computational complexity in the background correction of long-term time-lapse movies of many frames (details are provided in Supplementary Note 3). With the

estimation of $S(x)$, $D(x)$ and $B_i$, the correction of a time-lapse movie will be:

$$I_i^{corr}(x) = \left(I_i^{meas}(x) - D(x)\right)/S(x) - B_i + B_{norm}, \qquad (9)$$

where $B_{norm}$ is an arbitrarily chosen background. In practices, we set $B_{norm} = \text{mean}_i(B_i)$ for bright-field movies to ensure that the BaSiC-corrected movie is in the same intensity range as the raw movie. As for fluorescence movies, we can use $B_{norm} = 0$ to remove the background signal.

**Microscopy data sets.** 45 WSIs of four types of specimens used often in biological investigations were collected using a Nikon Ti microscope: four fluorescence WSIs of mouse brain sections, three fluorescence WSIs of mouse kidney sections (Molecular Probes FluoCells Prepared Slide #3), four fluorescence WSIs of tissue culture cells (Molecular Probes FluoCells Prepared Slide #2), and four bright-field WSIs of H&E stained tissue section, each in three different channels. Images were acquired with a $\times 10/0.45$ NA Plan Apochromat objective using a Lumencor SpectraX light source for excitation and emission filters for DAPI, Fluorescein and Cy3. The microscope was controlled by μManager[15]. Each WSI contains 50–200 image tiles and is stitched using Grid-wise stitching Plugin in Fiji[16] before and after intensity collection. For each data set, we also acquire reference images for the flat-field $S(x)$ using a concentrated fluorescent dye solution (Supplementary Note 1), and the dark-field $D(x)$ with no light entering the camera. For brain and cell specimens at Cy3 channel, we observed an additive light in the image background, which can be due to residual excitation light or stray light but not captured in the dark-field calibration. Details of all WSIs and the corresponding corrections are included in Supplementary Figs 5–8.

Besides the above WSIs, the microscopy data sets used in this study also include: (i) 6,000 synthetic images with three different levels of cell density (Supplementary Note 6); (ii) The image collection used in CIDRE (ref. 5) including 10 real microscopy data sets, one photography data set and one synthetic microscopy data set; (iii) One long-term ($\sim 6$ days) time-lapse movie (one bright-field and three fluorescence channels) of hematopoietic stem and progenitor cells (Hoppe *et al.*[12]), which contain $\sim 7,000$ bright-field frames (acquired every 2 min) and $\sim 320$ fluorescence frames for each fluorescence channel (acquired every 30 min).

**Baseline shading correction methods.** We compare BaSiC to three state-of-the-art retrospective methods and three prospective methods for shading correction. Each method is summarized below:

CIDRE (ref. 5) is recently published state-of-the-art illumination correction method[5], as it achieves the best performance among 13 shading correction methods. CIDRE can estimate both $S(x)$ and $D(x)$ and hence does not require any reference images. Yet the simultaneous estimation of two unknown parameters is not stable when the available images are limited or when images are corrupted with 'spike'-like noise.

The background correction module in the software package CellProfiler[7] approximates $S(x)$ using the mean image intensity computed at every location and subsequently smoothed by a median filter of a user defined kernel size (we use a kernel size of 20% of the image size). $D(x)$ is neglected in CellProfiler.

Coster *et al.*[6] proposed a shading correction method for high-throughput microscopy[6]. It first subtracts $D(x)$, required to be obtained via a prospective method, from all images and approximates $S(x)$ using the median image intensity computed at every location after subtraction. In our implementation, we further smooth $S(x)$ with a median filter (kernel size is 20% of the image size, which is found to be optimal). Strictly speaking, Coster is not a complete retrospective method, as the calibration of $D(x)$ is needed for correction.

Calib-zero[5] is a prospective method which approximates the flat-field $S(x)$ as the average of images of a plastic fluorescent reference slide[17]. The dark-field $D(x)$ is modelled by averaging images with the shutter closed or the light source turned off or otherwise blocked. The major drawback of this method is that the reference slides often have a different thickness as compared to real histology slides, which makes the approximation of $S(x)$ inaccurate (Supplementary Note 1).

Empty-zero[5] is another prospective method that approximates the flat-field $S(x)$ as the average of empty images taken at various locations[18]. The calibration of the dark-field $D(x)$ is same as in Calib-zero. This method is appropriate for bright-field images or fluorescence images when the medium fluoresces[5] but will be not applicable for images without a medium. Both the correction of Calib-zero and Empty-zero are obtained from Smith *et al.*[5] alongside the data.

Concentrated dye solution approximates the flat-field $S$ using images of a thin layer of concentrated dye[4]. The calibration of the dark-field $D(x)$ is same as in the above two prospective approaches. This method is usually more accurate than Calib-zero as it has a similar thickness to real specimens (Supplementary Note 1). In our study, we use the concentrated dye solution as the ground-truth to evaluate our correction of WSI.

**Evaluation protocol.** For synthetic data, where the ground-truth $S(x)$, $D(x)$, and the true shading-free images, $I^{true}(x)$, are available, we quantify the error of the estimated flat-field, $S^{est}(x)$, the estimated dark-field, $D^{est}(x)$, and the corrected image, $I^{corr}(x)$ with a score, $\Gamma$, defined as the mean absolute deviation between estimation/correction and the ground-truth, normalized by a baseline difference

(the baseline for $S(x)$, $D(x)$, and $I^{corr}(x)$ is a uniform flat-field, zero dark-field and uncorrected image $I^{meas}(x)$, respectively):

$$\Gamma(S^{est}) = \frac{\sum_x |S^{est}(x) - S(x)|}{\sum_x |1 - S(x)|}, \quad \Gamma(D^{est}) = \frac{\sum_x |D^{est}(x) - D(x)|}{\sum_x |D(x)|},$$

$$\Gamma(I^{corr}) = \frac{\sum_x |I^{corr}(x) - I^{true}(x)|}{\sum_x |I^{meas}(x) - I^{true}(x)|}$$

The score $\Gamma$ is in the interval $[0, \infty)$, where 0 indicates perfect estimation/correction, 1 indicates the same amount of error as the uncorrected images, and $>1$ indicates greater disagreement.

For real microscopy images where the ground truth is not available, a thorough evaluation of a shading correction method is not trivial. There are generally two different approaches in the literature to measure the shading correction quality. The first approach uses the prospectively obtained $S$ as a reference and quantifies the error to a retrospectively estimated $S(x)$ (for example in Coster *et al.*[6]). One key challenge of such a validation is how to acquire a reliable reference $S(x)$. Generally, identical microscope settings, similar specimen thickness, and averaging of multiple reference images (or taking the median to be robust to outliers) improve the accuracy of the reference. In our study, we examine the reliability of our prospectively obtained $S(x)$ by inspecting the smoothness of the shading-corrected WSI using $S(x)$. Taking the reference $S(x)$ as the ground-truth, we can quantitatively measure the performance of a retrospective method using the estimation score, $\Gamma(S^{est})$. The second type of evaluation is based on a correction score[5]. In real images where the ground-truth, shading-free images are unavailable, the correction score, $\Gamma'(I^{corr})$, is based on the absolute difference between pairs of overlapping, corrected images $I_a^{corr}(x)$ and $I_b^{corr}(x)$ that are precisely aligned, normalized by the benchmark error of the uncorrected image pairs, $I_a^{meas}(x)$ and $I_b^{meas}(x)$, as:

$$\Gamma'(I^{corr}) = \frac{\sum_x |I_a^{corr}(x) - I_b^{corr}(x)|}{\sum_x |I_a^{meas}(x) - I_b^{meas}(x)|} \qquad (10)$$

This approach avoids the collection of reference images, yet a perfect correction for real images is impossible, as the disagreement between overlapping images includes many other sources besides uneven illumination, such as noise, alignment error, and photobleaching (the second image, $I_b^{meas}(x)$, of the image pair usually has a lower signal than the first one, $I_a^{meas}(x)$, due to the bleaching of fluorescence dyes even without shading). In CIDRE (ref. 5), an extra intensity normalization process is involved, which normalizes the median and the s.d. of image pair before and after correction, that is, $I_a^{corr}(x)$, $I_b^{corr}(x)$, $I_b^{meas}(x)$ to the reference $I_a^{meas}(x)$. However, we do not include any extra normalization process in our study, as an intensity normalization between the uncorrected pairs will affect the assessment of shading correction. Nevertheless, the scores we obtained without normalization (Fig. 2a, Supplementary Fig. 3) are very similar to those reported in CIDRE (ref. 5).

**Data and software availability.** BaSiC is available as a Fiji/ImageJ Plugin from the Fiji/ImageJ update site http://sites.imagej.net/BaSiC/ and from our software website https://www.helmholtz-muenchen.de/icb/research/groups/quantitative-single-cell-dynamics/software/basic/index.html, where we also provide five different microscopy data sets used in the manuscript to demonstrate the usage of BaSiC. See also Supplementary Note 7 for installation, usage details and practical tips.

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

## Acknowledgements

Whole slide imaging data was acquired at the Nikon Imaging Center at the University of California, San Francisco. T.P. acknowledges a Humboldt Postdoctoral Research Fellowship. T.S. acknowledges financial support for this project from the SNF and SystemsX.ch. C.M. acknowledges support from the Deutsche Forschungsgemeinschaft (MA 5282/3-1). F.T. acknowledges funding from the Bayerisches Forschungsnetzwerk BioSysNet. T.P. and N.N. acknowledge the support of the Collaborative Research Centre SFB 824 (Z2). The authors acknowledge Kevin Smith to provide the microscopy image collection used in CIDRE[5] and thank Michael K. Strasser, Felix Buggenthin, Maximilian Baust and Sailesh Conjeti for insightful discussion.

## Author contributions

T.P. developed and implemented BaSiC. L.W. contributed to method development. K.T. provided WSI data. T.S. provided time-lapse data. T.P. and C.M. wrote the manuscript. F.J.T., K.T. and N.N. commented on the method and the manuscript. C.M. and N.N. supervised the project. All authors approved the final draft.

## Additional information

**Competing interests:** T.P., N.N. and C.M. have applied for a patent relating to this work.

**Publisher's note**: 

