## [Peer Review File · Nature Communications]

Reviewers' comments:

Reviewer #1, expert in image analysis algorithms (Remarks to the Author):

Summary of Key results:

The authors present a retrospective method for simultaneously correcting the both additive effects (dark current/offset/background) and spatially variable multiplicative effects (illumination/gain variability/"shading") in light microscopy images. The authors also claim to "solve" photobleaching, though it is not clear that this is the case (I address this below).

This method is based on applying regularized low rank and sparse matrix decomposition to batches of images. The low rank component of the decomposition comprises the illumination variability ("shading") and additive components, where the rank is low because the additive and multiplicative components are assumed to be common between the images, sorted versions of which form columns of the measurement matrix being decomposed. The sparse component accounts for residuals to this decomposition. Regularization is applied to impose spatial smoothness on both the additive and multiplicative components. The objective function for determining the decomposition is reformulated to allow practical evaluation and reliable convergence.

The method performs well in comparison to competing retrospective methods, requiring only a small number of images. The authors convincingly demonstrate this performance advantage across a wide variety of images.

Originality, interest and significance:

Originality: While sparse+low-rank decomposition has been applied before within the computer vision field to correct for undesirable variability within image collections (see e.g. RASL-it would be good to cite this or similar papers), to my knowledge this is the first time it has been applied to light microscopy images. This work therefore represents a novel approach to correcting optical microscopy aberrations.

Significance: Illumination variability affects nearly all light microscopy data and can introduce dramatic artifacts (easily a C.V of 20-30% or more) in any fluorescence quantification that is performed on these images. While I would disagree that prospective correction is "laborious" in most experiments, it does require some effort and this effort is apparently too much for many bioimaging researchers - all too often this effect is not corrected for. Furthermore, for some samples in in some imaging conditions (as the authors demonstrate convincingly in) it is difficult or impossible to acquire accurate correction images for prospective correction. Competing retrospective methods require larger numbers of images to reach similar accuracies. This work therefore represents a significant contribution to the bioimaging field.

Interest: Any researcher who is acquiring or has acquired images which comply with the assumptions inherent to this correction method (I address these assumptions more below) and who wants to either quantify intensities or create mosaic (stitched) images would be interested in this method. While researchers only wishing to count or measure the morphology of objects within optical microscopy images could in theory omit these corrections if their analysis algorithms are locally adaptive, even in these cases correcting for variable illumination and background can ease analysis.

Data and methodology:

The methods presented and their applications are generally sound, with the exception of photobleaching correction. The data to which the method is applied is perhaps not perfectly representative, but is diverse and the results are convincing.

Photobleaching: My primary concern with the methodology is in regards to photobleaching correction. Each time photobleaching is mentioned within the methods and model descriptions it is in reference to an additive term, e.g. line 54 referring to eqtn. 2 and again in the supplement. However, photobleaching is analogous to a decay process, with a fractional decrease in fluorophore populations per unit time. The correction for photobleaching is therefore multiplicative, whereas my understanding from the text is that the author's bleaching correction consists of allowing the Bi offset to vary across images, which would correct only for bleaching of uniform background

fluorescence in the image and not the rescaling of foreground sample fluorescence that occurs due to photobleaching. If the authors are applying a multiplicative correction for (other than for illumination) it is not clearly explained in the methods description.

Photobleach correction based on an accurate and justified exponential decay model of the photobleaching process is already available as a plugin in Fiji/ImageJ. For data where this model is not applicable, it is not clear that the method presented here is superior to per-image normalization/histogram equalization. I am therefore not currently convinced that photobleaching correction is adequately addressed by this method, nor should it play a prominent role in the manuscript title/abstract. However, the corrections the method presented do provide (illumination/flat-field correction, variable background/offset) are useful and still worthy of publication.

Assumptions of image properties: My second concern is the assumption inherent to this method that the foreground objects are uncorrelated within the image batch being corrected. The authors acknowledge this assumption, but it is addressed only at the end of the supplement. While many image sets meet this assumption, for the sake of the readers of this journal this assumption should be more prominently addressed within the manuscript, to avoid application of this method to inappropriate data. For example, in many higher-magnification, imaging experiments a single cell is present in the field of view and the microscopist usually centers this cell within the field of view. In such an experiment the method presented here would interpret the consistently higher image intensities within the center of the field of view as a local increase in $S(x)$, and the resulting correction would then be incorrect, causing the removal of true fluorescence variability. This same issue will affect all of the retrospective methods that the current method is compared to. I therefore do not think that this is a fatal flaw, but rather just that this risk should be more clearly communicated to the reader.

Similarly, it would be useful to mention in a more prominent location the required fraction of image area which must be background for the baseline subtraction method to be accurate.

Appropriate use of statistics and uncertainties: I did not identify any issues with the statistics applied or uncertainties presented. The supplemental method relating the correction method to statistics of large image collections was helpful.

Conclusions: I do not disagree with the conclusions presented, with the exception of the conclusion that the method corrects for photobleaching.

Suggested improvements:

As suggested above, decreasing the emphasis on photobleaching correction and more prominently clarifying the assumptions regarding the nature of the image content would make SLIC a resource which is more likely to be appropriately applied.

I also think it would be very useful to discuss more within the text what aspects of different image data affect the number of images required for accurate correction with SLIC. In figure S3 different datasets require anywhere from 5 to 100 images in order to produce a correction that is equivalent to prospective methods. What is it specifically about these image data that caused this variability? Confluency? Variability in object positioning within the image? Uniformity of foreground fluorescence? If the reader has a better sense of these factors they will be better able to decide how many images of their specific sample will be required to apply SLIC.

Other minor (optional) improvements I would suggest include:

Use the term "flat field correction" once or more within the text when referring to the $S(x)$ /shading, so that people who are more accustomed to this term rather than shading will still find the paper/tool when they search.

Provide an update site for the Fiji plugin rather than a download.

One sentence explaining why this method is superior (it is!) to local contrast equalization methods (e.g. CLAHE) - these methods are often applied when a correction such as SLIC should be applied instead.

Clarify the meaning of "empirically set by bootstrapping using simulations" in the supplement when referring to the denominators of the terms used to set the weights on the regularization parameters.

Discuss the computational complexity (how SLIC execution time scales with number of images and image size).

Lots of typos (mainly in the supplement): In the main text in equation 4 you use regular $F(*)$ when you should use a "fancy" F to indicate fourier transform, as used in the supplement. This is confusing because you also use F for foreground above. Also, several "spell check doesn't catch them typos like "constraint optimisation", "gradient decent" S1 you say "outliner" instead of "outlier", etc.

References:

It seems appropriate to reference one or more previous computer vision / image processing applications of sparse+low rank decomposition. Otherwise referencing was fine.

Clarity and context: This was fine overall (with the exception of the issues mentioned above). The methods description could perhaps use an example of the specific elements of the measurement matrix, i.e. where the upper / upper left elements are shown with ellipses, illustrating the final composition of this matrix. The authors provide enough information for this to be inferred, but it takes some careful reading. Otherwise good.

Summary: SLIC is a useful and novel tool which is overall clearly and convincingly presented and, with some revisions, worthy of publication in nature communications.

Reviewer #2, expert in image processing (Remarks to the Author):

This paper can be the state-of-the-art. The topic 'illumination correction' is not very exciting for natural sciences research community (saying it such that I work on it several years ago) but solves a very serious and acute problem of optical microscopy. My general impression of this paper was positive with the following major comments:

- The method is well grounded, current, elegant mathematics and present impressive results.

- I did not like that the paper is very close to CIDRE (Smith et al.), probably a bit closer than necessary (wording, data, main idea), but I think it is still acceptable especially because of the impressive improvement in terms of results. I would appreciate if Authors better cite and highlight similarities and differences.

- Name is very unlucky, SLIC superpixels with 2100 citations (<http://ivrl.epfl.ch/research/superpixels>) is a very popular method similarly in image analysis. This will be hard for the spread of the current method and confusing for the users of both approaches. I would really encourage Authors to change it.

Minor comments:

line 30-31: This sentence is a bit weak to start. I think you should either talk about optical microscopy right away or put other imaging techniques eg. electron microscopy on the list.

line 43-46: I think one should be very careful here with defining image formation. It is absolutely misleading to think that image formation has a constant and linear scaling factor. There are several nonlinear and interconnected components that are neglected in most of the correction algorithms. I think Authors may say here that this model is an approximation. Of course it is a

good approximation and the method will work in practice with high accuracy.

Interestingly one of these factors that is not modeled by the linear model correctly is near the upper sensitivity of the camera (saturation intensity). Cameras lose their linearity and plots such as the one in Supp Fig 1A become nonlinear near the saturation region.

line 62: Smith et al released Cidre as a set of CellProfiler plugins as well, not only ImageJ.

line 70-88: As said above I support this paper as it brings more stable results than any previous methods, but the basic minimization idea that Authors present (smoothness and fitting criteria and assuming D is stable) is exactly the 3 criteria described and minimized by Smith et al. I think here the major differences should be mentioned/highlighted (that is the fast and elegant solution) and clearly demonstrate that there is a great add-on on the top of earlier work in terms of methodology.

line 104: Write a few words about (Calib-zero and Empty-zero), these were technical abbreviations in Smith et al. Probably that could go to page 7.

line 107-111: these sentences contradict. First you say SLIC works on less images better than other methods; second you mention this is practically relevant in HCS because thousands of images are available?!

line 146: Instead of high-throughput I would use high-content all around the paper. High-throughput is more commonly used for plate-readers than microscopes

Runtime and memory reqs. Please describe the time and memory consumption of the method in terms of number and size of images and difficulty of images.

Supplementary Note 4.: empirical values 2000 and 800 as normalization factors. I think it is great that the method is parameter-free. But if you use empirical parameter settings, I would prefer to optimize them on a large representative dataset. These values seem very round and hand-picked, I think they can be further optimized.

References are not unified in the Supplementary notes. (eg. [4] M A Model, [5] Michael Model).

In general I think the work represents high-quality theory and good solution. If authors address my comments I will happily support its publication.

Reviewer #1, expert in image analysis algorithms (Remarks to the Author):

Summary of Key results:

The authors present a retrospective method for simultaneously correcting the both additive effects (dark current/offset/background) and spatially variable multiplicative effects (illumination/gain variability/"shading") in light microscopy images. The authors also claim to "solve" photobleaching, though it is not clear that this is the case (I address this below).

This method is based on applying regularized low rank and sparse matrix decomposition to batches of images. The low rank component of the decomposition comprises the illumination variability ("shading") and additive components, where the rank is low because the additive and multiplicative components are assumed to be common between the images, sorted versions of which form columns of the measurement matrix being decomposed. The sparse component accounts for residuals to this decomposition. Regularization is applied to impose spatial smoothness on both the additive and multiplicative components. The objective function for determining the decomposition is reformulated to allow practical evaluation and reliable convergence.

The method performs well in comparison to competing retrospective methods, requiring only a small number of images. The authors convincingly demonstrate this performance advantage across a wide variety of images.

Originality, interest and significance:

Originality: While sparse+low-rank decomposition has been applied before within the computer vision field to correct for undesirable variability within image collections (see e.g. RASL-it would be good to cite this or similar papers), to my knowledge this is the first time it has been applied to light microscopy images. This work therefore represents a novel approach to correcting optical microscopy aberrations.

We thank the Reviewer for the positive comments. We now cite the RASL paper alongside Robust PCA in the revised version of our manuscript (line 75-78):

"The optimisation is solved in an iterative fashion using the linearized augmented Lagrangian method⁸, which is widely used in sparse matrix decomposition like Robust PCA⁹ and RASL¹⁰"

Significance: Illumination variability affects nearly all light microscopy data and can introduce dramatic artifacts (easily a C.V of 20-30% or more) in any fluorescence quantification that is performed on these images. While I would disagree that prospective correction is "laborious" in most experiments, it does require some effort and this effort is apparently too much for many bioimaging researchers - all too often this effect is not corrected for. Furthermore, for some samples in some imaging conditions (as the authors demonstrate convincingly in) it is difficult or impossible to acquire accurate correction images for prospective correction. Competing retrospective methods require larger numbers of images to reach similar accuracies. This work therefore represents a significant contribution to the bioimaging field.

We agree that prospective correction may not always be "laborious". However, this extra effort, as pointed out by the Reviewer, is often too much for many bioimaging researchers (about half of researchers do not use any flat-field correction according to the survey of Smith et al. 2015).

In the revised version of our manuscript, we replaced "laborious" with "extra".

Interest: Any researcher who is acquiring or has acquired images which comply with the assumptions inherent to this correction method (I address these assumptions more below) and who wants to either quantify intensities or create mosaic (stitched) images would be interested in this method. While researchers only wishing to count or measure the morphology of objects within optical microscopy images could in theory omit these corrections if their analysis algorithms are locally adaptive, even in these cases correcting for variable illumination and background can ease analysis.

We thank the Reviewer for these comments and incorporate them in first paragraph of the updated discussion to make the paper more targeted to potential users (line 148-157):

“BaSiC will have immediate attraction to researchers who create stitched images, since correcting uneven illumination improves stitching and mosaic image quality. Besides, BaSiC can be also used as a pre-processing step in conjunction with automatic methods such as cell counting or measuring the morphology of cells and thus improving down-stream analysis. The crucial contribution of BaSiC is to improve intensity quantification in both static and time-lapse imaging data. Unlike local contrast equalization methods, which could distort the true intensity variations within an original image or across multiple images, BaSiC is built on solid physical models of optical imaging and hence is able to recover biologically relevant intensities for image quantification.”

Data and methodology:

The methods presented and their applications are generally sound, with the exception of photobleaching correction. The data to which the method is applied is perhaps not perfectly representative, but is diverse and the results are convincing.

Photobleaching: My primary concern with the methodology is in regards to photobleaching correction. Each time photobleaching is mentioned within the methods and model descriptions it is in reference to an additive term, e.g. line 54 referring to eqtn. 2 and again in the supplement. However, photobleaching is analogous to a decay process, with a fractional decrease in fluorophore populations per unit time. The correction for photobleaching is therefore multiplicative, whereas my understanding from the text is that the author's bleaching correction consists of allowing the B_i offset to vary across images, which would correct only for bleaching of uniform background fluorescence in the image and not the rescaling of foreground sample fluorescence that occurs due to photobleaching. If the authors are applying a multiplicative correction for (other than for illumination) it is not clearly explained in the methods description.

Photobleach correction based on an accurate and justified exponential decay model of the photobleaching process is already available as a plugin in Fiji/ImageJ. For data where this model is not applicable, it is not clear that the method presented here is superior to per-image normalization/histogram equalization. I am therefore not currently convinced that photobleaching correction is adequately addressed by this method, nor should it play a prominent role in the manuscript title/abstract. However, the corrections the method presented do provide (illumination/flat-field correction, variable background/offset) are useful and still worthy of publication.

We agree that the photobleaching effect in the foreground fluorophores is multiplicative, while the additive background term B_i used in our method only accounts for the bleaching in the background fluorescence. As pointed out by the Reviewer, this distinction was not well described in our initial manuscript. In the revised manuscript, we added a paragraph to clarify the strengths and limitations of our method in terms of foreground and background photobleaching (line 187-198):

“Although BaSiC can compensate background variation, no matter if it is caused by bleaching or by switching microscopy settings, it does not account for variation in the foreground sample fluorescence that may also occur due to photobleaching. In the presented long-term single-cell time-lapse measurements, the dominant

corrupting factor is the background variation caused by medium bleaching. Hence subtraction of background bleaching greatly improves the intensity quantification of single cells (as shown in Fig. 3). In fact, existing photobleaching correction methods (such as the Bleaching Correction Plugin in Fiji/ImageJ) are not suitable for correcting foreground cell bleaching in our movies: these methods either assume constant intensity or stable intensity distribution of each frame, which is certainly not the case for transcription factor expression during cell differentiation, where the signal varies depending on the cell type and time.”

From our perspective, correcting background variation (no matter if it is caused by bleaching or by switching microscopy settings) as provided by our method is an important contribution especially for long-term single-cell imaging (see e.g. Etzrodt, Endeke & Schroeder 2014). To the best of our knowledge, our method is the only one among all the available multi-image based retrospective methods able to do this. In accordance to the Reviewer’s comment, we updated the title of our manuscript to ‘BaSiC: A Tool for Background and Shading Correction of Optical Microscopy Images’ and now mention the bleaching correction in the abstract and main text, where we tried to be more specific by writing “correction of background bleaching” or “correction of background variation”.

Assumptions of image properties: My second concern is the assumption inherent to this method that the foreground objects are uncorrelated within the image batch being corrected. The authors acknowledge this assumption, but it is addressed only at the end of the supplement. While many image sets meet this assumption, for the sake of the readers of this journal this assumption should be more prominently addressed within the manuscript, to avoid application of this method to inappropriate data. For example, in many higher-magnification, imaging experiments a single cell is present in the field of view and the microscopist usually centers this cell within the field of view. In such an experiment the method presented here would interpret the consistently higher image intensities within the center of the field of view as a local increase in $S(x)$, and the resulting correction would then be incorrect, causing the removal of true fluorescence variability. This same issue will affect all of the retrospective methods that the current method is compared to. I therefore do not think that this is a fatal flaw, but rather just that this risk should be more clearly communicated to the reader.

We agree with the Reviewer that we should discuss this assumption at a more prominent position in the paper. Hence, we added one paragraph to the discussion of the revised manuscript (line 168-186), where we now write along the Reviewer’s suggestion:

“As any shading correction method, BaSiC has limitations. One key assumption of BaSiC and all other previously mentioned multi-image based retrospective methods is that the foreground of every image to be processed should be uncorrelated with the foreground of every other image. This assumption can be violated for time-lapse movies of static and quasi-static objects, e.g. for a single cell of high magnification that is always in the centre of the field of view. In such cases, BaSiC would consider the consistently higher image intensities in the centre of the field of view as a local increase in S , causing removal of the true fluorescence variability. Nevertheless, in practice, BaSiC still has some tolerance to correlation, e.g. it performs well in a movie of proliferating and slowly moving embryonic stem cell colonies (as shown in Supplementary Video 2), in which consecutive frames are correlated. Meanwhile, the regularisation parameter λ_s can be used to tune the resulting model so that it is more suitable for more correlated images. Larger values of λ_s lead to a smoother estimation of the low rank component, thus rejecting smaller static objects included in the estimated S . Another practically useful strategy is to take samples from the movie with a large time gap in between to make images less correlated. In any case, we suggest users to visually inspect the estimated shading profiles before making a correction in such challenging cases: smooth S usually indicates a good shading correction, while local inhomogeneities that come from highly corrected foreground objects are a hint of nonoptimal correction.”

Similarly, it would be useful to mention in a more prominent location the required fraction of image are which must be background for the baseline subtraction method to be accurate.

Following the Reviewer's suggestion, we added one sentence in the discussion of the revised manuscript (line 198-204), where we write:

"It should also be noted for fluorescence images, the estimated baseline can converge to the foreground, when the foreground fraction of an image is higher as 50%. This does not affect the practical usage of BaSiC when a high cell density is reached only at the end of a movie. Typically then, the bleaching effect is already weak (bleaching usually decays exponentially) and hence the correction for those frames can be skipped. By contrast, for bright-field images, BaSiC is robust to different levels of cell density in background correction."

Appropriate use of statistics and uncertainties: I did not identify any issues with the statistics applied or uncertainties presented. The supplemental method relating the correction method to statistics of large image collections was helpful.

Conclusions: I do not disagree with the conclusions presented, with the exception of the conclusion that the method corrects for photobleaching.

Suggested improvements:

As suggested above, decreasing the emphasis on photobleaching correction and more prominently clarifying the assumptions regarding the nature of the image content would make SLIC a resource which is more likely to be appropriately applied.

We thank the Reviewer for this suggestion. We now discuss more clearly that our tool corrects for photobleaching of the background only (see above).

I also think it would be very useful to discuss more within the text what aspects of different image data affect the number of images required for accurate correction with SLIC. In figure S3 different datasets require anywhere from 5 to 100 images in order to produce a correction that is equivalent to prospective methods. What is it specifically about these image data that caused this variability? Confluency? Variability in object positioning within the image? Uniformity of foreground fluorescence? If the reader has a better sense of these factors they will be better able to decide how many images of their specific sample will be required to apply SLIC.

We agree that the discussion of the number of images required by BaSiC provides a useful practical tip for readers. Accordingly we added the following paragraph in Supplementary Note 7:

"In practice, the required number of images for BaSiC depends on many factors, such as confluency, thickness of background medium and uniformity of foreground fluorescence. The required image number is smallest for cells of low confluency and a thick medium (such as MICROFLUID dataset in Supp. Fig. 3). For images without any background signal (such as confocal images or cells without medium, e.g. HIST-CONFOCAL and HCS-ACTIN, Supp. Fig. 3), estimation has to rely on the foreground alone, and hence uniformity of foreground fluorescence will play an important role: a more uniform foreground requires fewer images."

Other minor (optional) improvements I would suggest include:

Use the term "flat field correction" once or more within the text when referring to the $S(x)$ /shading, so that people who are more accustomed to this term rather than shading will still find the paper/tool when they search.

Following the Reviewer's comment, we changed our term for $S(x)$ to "flat-field" and for $D(x)$ to "dark-field" throughout the revised manuscript. E.g. in line 45-47, we now write:

"... the multiplicative term $S(x)$ (known as flat-field). The additive term, $D(x)$, is dominated by camera offset and thermal noise, which are present even if no light is incident on the sensor (known as dark-field)."

Provide an update site for the Fiji plugin rather than a download.

As suggested by the Reviewer, we now provide a website for BaSiC and its future updates at <https://www.helmholtz-muenchen.de/icb/research/groups/quantitative-single-cell-dynamics/software/BaSiC/index.html>. Here, we present a download link for BaSiC, install instructions, the demo examples used in our manuscript and two video tutorials that detail the use of BaSiC for static images and time-lapse data.

One sentence explaining why this method is superior (it is!) to local contrast equalization methods (e.g. CLAHE) - these methods are often applied when a correction such as SLIC should be applied instead.

We thank the Reviewer for this insightful suggestion. We added a sentence in the first paragraph in the updated manuscript to explain the essential difference between BaSiC and local contrast equalization methods (line 152-157):

"The crucial contribution of BaSiC is to improve intensity quantification in both static and time-lapse imaging data. Unlike local contrast equalization methods, which could distort the true intensity variations within an original image or across multiple images, BaSiC is built on solid physical models of optical imaging and hence is able to recover biologically relevant intensities for image quantification."

Clarify the meaning of "empirically set by bootstrapping using simulations" in the supplement when referring to the denominators of the terms used to set the weights on the regularization parameters.

We agree that "empirically set by bootstrapping using simulations" was indeed too vague to clarify the process to set the denominators. In the updated Supplementary Note 4, we write in a more specific way:

"The denominators, 800 and 2000, are empirically determined using synthetic data, where we try to minimise the errors between our repeatedly estimated S and D from bootstrapping samples and the ground-truth S and D (generation of synthetic images of different cell densities is described in Supplementary Note 6). In practice, we find that the performance of BaSiC is robust to a wide range of denominators and the error is minimal for the chosen numbers."

Discuss the computational complexity (how SLIC execution time scales with number of images and image size).

In general, BaSiC computation is fast. E.g., it needs less than 1 min to process 100 images. Its execution time is independent of image size (as we downsample any image to 128x128) but does linearly scale with number of images. In our old implementation, we computed S , D and per-image B_i at the same time. This was not very efficient in terms of computational speed and memory usage for large image sequences with many frames. Since we need B_i for every frame to correct baseline drift, we have to decompose a large measurement matrix whose column number is twice of the image number, since the matrix is composed of

both original and sorted images. In our updated implementation, we propose a two-step strategy to estimate B_i ; the first step is to compute S and D only using the matrix with sorted intensities; in the second step we estimate B_i using the unsorted measurement matrix and the estimated S and D from step 1 as inputs. In comparison to the previous, simultaneous estimation of S , D and B_i , computing B_i with S and D as inputs is much faster due to a reduced degree of freedom.

The Method section of the main text and Supplementary Note 3 have both been updated accordingly. Also, the two step procedure is demonstrated using an exemplary application in the readme.txt coming with the package and on our website. We also added two paragraphs to Supplementary Note 7 to explain the complexity of BaSiC and to discuss potential strategies to accelerate the computation when dealing with image sequences with large number of frames:

"In general, BaSiC computation is fast: it usually processes hundreds of images within minutes at a standard laptop. A key factor for such a fast computation is that we always rescale the input images to 128x128 pixels. This resizing saves computational expense without losing accuracy since S and D are usually smooth and B_i is a scalar. However, the computational complexity of BaSiC linearly increases with the number of images increases, as the number of columns of the matrix to be decomposed is equivalent to the number of images."

"BaSiC also provides an option to separate shading correction (estimation of S and D) and background correction (estimation of B_i). This provides a very effective strategy to process long-term time-lapse movies with thousands of frames: firstly we can compute S and D using a small image subset, which is much faster than the computation using the whole image sequence; in the second step we use the precomputed S and D as inputs and compute per-image B_i in a block-wise fashion (i.e. we can divide a long image sequence into several short sequences and compute B_i for each short sequence either sequentially or in parallel). Hence, this implementation optimises both the computational time and the memory consumption."

In the main text, we added one sentence in the discussion to stress BaSiC's quick computation (line 157-159):

"Besides being accurate, BaSiC is also fast to compute: in our Fiji implementation, it usually processes hundreds of images (regardless of the original image size) within minutes on a standard laptop."

Lots of typos (mainly in the supplement): In the main text in equation 4 you use regular F^* when you should use a "fancy" F to indicate fourier transform, as used in the supplement. This is confusing because you also use F for foreground above. Also, several "spell check doesn't catch them typos like "constraint optimisation", "gradient descent" S1 you say "outliner" instead of "outlier", etc.

We are sorry about the typos in the supplement. We have proofread it again and corrected these typos. We also use "fancy" F to represent Fourier Transform in Equation 4 in the updated manuscript.

References:

It seems appropriate to reference one or more previous computer vision / image processing applications of sparse+low rank decomposition. Otherwise referencing was fine.

In the updated manuscript, RASL is added alongside Robust PCA as exemplary literature in sparse and low rank decomposition.

Clarity and context: This was fine overall (with the exception of the issues mentioned above). The methods description could perhaps use an example of the specific elements of the measurement matrix, i.e. where the upper / upper left elements are shown with ellipses, illustrating the final composition of this matrix. The authors provide enough information for this to be inferred, but it takes some careful reading. Otherwise good.

We thank the Reviewer for this suggestion. In the revised manuscript, we denote the low rank component of the matrix as I^B , and the sparse part as I^R . This nomenclature is now used both in Methods and Figure 1.

Summary: SLIC is a useful and novel tool which is overall clearly and convincingly presented and, with some revisions, worthy of publication in nature communications.

We thank the Reviewer again to acknowledge the strength of BaSiC. The revisions suggested by the Reviewer clearly improved our manuscript.

Reviewer #2, expert in image processing (Remarks to the Author):

This paper can be the state-of-the-art. The topic 'illumination correction' is not very exciting for natural sciences research community (saying it such that I work on it several years ago) but solves a very serious and acute problem of optical microscopy. My general impression of this paper was positive with the following major comments:

- The method is well grounded, current, elegant mathematics and present impressive results.

- I did not like that the paper is very close to CIDRE (Smith et al.), probably a bit closer than necessary (wording, data, main idea), but I think it is still acceptable especially because of the impressive improvement in terms of results. I would appreciate if Authors better cite and highlight similarities and differences.

As pointed out by the Reviewer, several important features of BaSiC are indeed similar to CIDRE, e.g. the simultaneous estimation of flat-field $S(x)$ and dark-field $D(x)$. We added one sentence to better emphasize the similarities and the differences (line 69-74):

"Inspired by Smith et al.⁶, we build our method on the shading model (Eq. 1) which accounts for the effect of both S and D . Such a full model is superior as compared to a partial model that only considers S ⁶. Besides, per-image baseline B_i , flat-field S and dark-field D are modelled as low rank matrices (rank ≤ 2), whilst the data fitting error is modelled as a sparse residual and is penalised with a reweighted L1-norm (Fig. 1c)."

Moreover, data from the CIDRE paper has been used in our manuscript, since we believe that this is the fairest approach to make a benchmark comparison. However, we also use real WSI and long-term time-lapse movies to evaluate the performance of our method, which clearly goes beyond the data used in the CIDRE paper. In our discussion, we added several sentences to highlight the key difference between BaSiC and CIDRE (line 160-167):

"From a methodological point of view, there are two key differences between BaSiC and the state-of-the-art shading correction tools that also model flat-field S and dark-field D . The first distinctive feature of BaSiC is the reweighted L1-norm error measure, which allows for a quicker convergence when dealing with a limited number of images and, more importantly, results in increased resistance to outliers in data such as noise or debris. Secondly, besides S and D , BaSiC also estimates a per-image baseline B_i , which accounts for varying background in time-lapse movies. This correction of background bleaching is a unique feature of BaSiC that CIDRE and other existing methods cannot provide."

In the updated version, we also removed the Supplementary Table, which was a summary of the 12 image collections used in Smith et al. Instead, we now guide the reader directly to Smith et al. for dataset details.

- Name is very unlucky, SLIC superpixels with 2100 citations (<http://ivrl.epfl.ch/research/superpixels>) is a very popular method similarly in image analysis. This will be hard for the spread of the current method and confusing for the users of both approaches. I would really encourage Authors to change it.

We thank the Reviewer for pointing this out. We changed the name of our tool and our title to "BaSiC: A Tool for Background and Shading Correction of Optical Microscopy Images". We hope this "BaSiC" tool will help biologists to solve the 'very serious and acute problem of optical microscopy', illumination correction, as mentioned earlier by the Reviewer.

Minor comments:

line 30-31: This sentence is a bit weak to start. I think you should either talk about optical microscopy right away or put other imaging techniques eg. electron microscopy on the list.

We thank the Reviewer for the suggestion. The updated manuscript starts directly with

“Optical imaging is an indispensable tool in biomedical research. [...]”.

line 43-46: I think one should be very careful here with defining image formation. It is absolutely misleading to think that image formation has a constant and linear scaling factor. There are several nonlinear and interconnected components that are neglected in most of the correction algorithms. I think Authors may say here that this model is an approximation. Of course it is a good approximation and the method will work in practice with high accuracy. Interestingly one of these factors that is not modeled by the linear model correctly is near the upper sensitivity of the camera (saturation intensity). Cameras lose their linearity and plots such as the one in Supp Fig 1A become nonlinear near the saturation region.

This is a very insightful point. We have changed “modelled” to “approximated” in the updated sentence to avoid confusion (line 41-43):

“The physical process of image formation can be approximated as a linear function³ that relates a measured image, $I^{meas}(x)$ at location x , to its uncorrupted true correspondence, $I^{true}(x)$, as [...]”

Moreover, we added one sentence in the Supplementary Note 1 to explain that our model does not account for nonlinear effects:

“However, it should be noted that although this linear model is generally accurate for shading correction, it does not account for all possible components in the image formation, e.g. nonlinear effects such as camera saturation are neglected in the model.”

line 62: Smith et al released Cidre as a set of CellProfiler plugins as well, not only ImageJ.

We have corrected the citation of CIDRE and now mention both the Fiji and CellProfiler plugin (line 59-61):

“A number of multi-image based approaches have been recently published, e.g. Smith et al.⁶ (Fiji and Cell profiler Plugin “CIDRE”), Coster et al.⁴ and Singh et al.⁷ (default module in CellProfiler)”

line 70-88: As said above I support this paper as it brings more stable results than any previous methods, but the SLIC minimization idea that Authors present (smoothness and fitting criteria and assuming D is stable) is exactly the 3 criteria described and minimized by Smith et al. I think here the major differences should be mentioned/highlighted (that is the fast and elegant solution) and clearly demonstrate that there is a great add-on on the top of earlier work in terms of methodology.

Following the Reviewer’s advice, we describe CIDRE more concretely and discuss the differences between CIDRE and BaSiC in more detail in the revised version of our manuscript.

line 104: Write a few words about (Calib-zero and Empty-zero), these were technical abbreviations in Smith et al. Probably that could go to page 7.

As suggested by the Reviewer, we added the explanation of these two methods in the Method section of the updated manuscript, alongside the method of concentrated dye solution, which is used to evaluate WSI correction (line 313-331)

- Calib-zero⁶ is a prospective method which approximates the flat-field S as the average of images of a plastic fluorescent reference slide¹⁵. The dark-field D is modelled by averaging images with the shutter closed or the light source turned off or otherwise blocked. The major drawback of this method is that the reference slides often have a different thickness as compared to real histology slides, which makes the approximation of S inaccurate (Supplementary Note 1).
- Empty-zero⁶ is another prospective method that approximates the flat-field S as the average of images of empty images taken at various locations¹⁶. The calibration of the dark-field D is same as in Calib-zero. This method is appropriate for bright-field images or fluorescence images when the medium fluoresces⁶ but will be not applicable for images without a medium. Both the correction of Calib-zero and Empty-zero are obtained from Smith et al.⁶ alongside the data.
- Concentrated dye solution approximate the flat-field S using images of a thin layer of concentrated dye⁵. The calibration of the dark-field D is same as in the above two prospective approaches. This method is usually more accurate than Calib-zero as it has a similar thickness to real specimens (Supplementary Note 1). In our study, we use the concentrated dye solution as the ground-truth to evaluate our correction of WSI.

line 107-111: these sentences contradict. First you say SLIC works on less images better than other methods; second you mention this is practically relevant in HCS because thousands of images are available?!

Although high-content screening can generate thousands of images, not every image shares the same shading profile. According to the study of Coster et al., images taken at the same position of each well have a similar S , which makes the number of images available for one shading estimation equivalent to the number of wells (96 or 384). In our revised manuscript, we explain this in more detail (line 106-109)

“This is practically relevant since WSI acquisition typically contains 50-200 tiles and high-content screening usually works with 96 and 384 well plates, where only images at the same position of each well share a shading profile - hence the number of images available for one estimation is 96 or 384.”

line146: Instead of high-throughput I would use high-content all around the paper. High-throughput is more commonly used for plate-readers than microscopes

We thank the Reviewer for the suggestion. In the updated manuscript, we use “high-content” to describe static images and “high-throughput” to describe time-lapse movie data for the sake of consistency.

Runtime and memory reqs. Please describe the time and memory consumption of the method in terms of number and size of images and difficulty of images.

We added two paragraphs to Supplementary Note 7 to explain the implementation of BaSiC and to discuss the computational complexity:

“In general, BaSiC computation is fast: it usually processes hundreds of images within minutes at a standard laptop. A key factor for such a fast computation is that we always rescale the input images to 128x128 pixels. This resizing saves computational expense without losing accuracy since S and D are usually smooth and B_i is a scalar. However, the computational complexity of BaSiC linearly increases with the number of images increases, as the number of columns of the matrix to be decomposed is equivalent to the number of images.”

“BaSiC also provides an option to separate shading correction (estimation of S and D) and background correction (estimation of B_i). This provides a very effective strategy to process long-term time-lapse movies with thousands of frames: firstly we can compute S and D using a small image subset, which is much faster than the computation using the whole image sequence; in the second step we use the precomputed S and D as inputs and compute per-image B_i in a block-wise fashion (i.e. we can divide a long image sequence into several short sequences and compute B_i for each short sequence either sequentially or in parallel). Hence, this implementation optimises both the computational time and the memory consumption.”

In the main text, we added one sentence in the discussion to stress its quick computation (line 157-159):

“Besides being accurate, BaSiC is also fast to compute: in our Fiji implementation, it usually processes hundreds of images (regardless of the original image size) within minutes at a standard laptop.”

Supplementary Note 4.: empirical values 2000 and 800 as normalization factors. I think it is great that the method is parameter-free. But if you use empirical parameter settings, I would prefer to optimize them on a large representative dataset. These values seem very round and hand picked, I think they can be further optimized.

We agree that "empirically set by bootstrapping using simulations" was indeed too vague to clarify the process to set the denominators. In the updated Supplementary Note 4, we write in a more specific way:

“The denominators, 800 and 2000, are empirically determined using synthetic data, where we try to minimise the errors between our repeatedly estimated S and D from bootstrapping samples and the ground-truth S and D (generation of synthetic images of different cell densities is described in Supplementary Note 6). In practice, we find that the performance of BaSiC is robust to a wide range of denominators and the error is minimal for the chosen numbers.”

References are not unified in the Supplementary notes. (eg. [4] M A Model, [5] Michael Model).

Thanks. We unified the references in the revised version of our manuscript.

In general I think the work represents high-quality theory and good solution. If authors address my comments I will happily support its publication.

We thank the Reviewer again for his helpful suggestions that clearly improved our tool and manuscript.

REVIEWERS' COMMENTS:

Reviewer #1 (Remarks to the Author):

The authors have addressed all of my salient concerns with the original manuscript. The tool they present (now cleverly renamed BaSiC) will be useful to the field and its improved efficiency and performance relative to existing methods will hopefully increase the number of bioimaging researchers who appropriately correct their light microscopy images prior to presentation and analysis, and increase reproducibility in the field as a result. I can now unreservedly recommend publication.

As a side note I would like to clarify one suggestion which I believe was misunderstood. It would likely increase the rate at which this tool is adopted within the field if the authors established an ImageJ/Fiji update site (rather than just providing a link to download the .jar), as described here: http://imagej.net/How_to_set_up_and_populate_an_update_site This not only makes installation easier, but also makes it very easy to distribute any updates and bug-fixes to your tool. This is, however, purely a suggestion and the current means of distribution are perfectly adequate.

Reviewer #2 (Remarks to the Author):

Authors addressed my concerns point by point. I accept the answers and agree with the changes in the manuscript. I support the publication of this work in Nature Communications and congratulate the Authors for the method and results that may put BASIC the state of the art.

Dear Reviewers,

We thank you for recommend to accept our manuscript. Below is point-to-point response for your requests.

Reviewer #1 (Remarks to the Author):

The authors have addressed all of my salient concerns with the original manuscript. The tool they present (now cleverly renamed BaSiC) will be useful to the field and it's improved efficiency and performance relative to existing methods will hopefully increase the number of bioimaging researchers who appropriately correct their light microscopy images prior to presentation and analysis, and increase reproducibility in the field as a result. I can now unreservedly recommend publication.

As a side note I would like to clarify one suggestion which I believe was misunderstood. It would likely increase the rate at which this tool is adopted within the field if the authors established an ImageJ/Fiji update site (rather than just providing a link to download the .jar), as described here: http://imagej.net/How_to_set_up_and_populate_an_update_site This not only makes installation easier, but also makes it very easy to distribute any updates and bug-fixes to your tool. This is, however, purely a suggestion and the current means of distribution are perfectly adequate.

We incorporated the reviewer #1's suggestion and now provides BaSiC as a update site in Fiji plugin list. We thank the reviewer for her/his valuable comments.

Reviewer #2 (Remarks to the Author):

Authors addressed my concerns point by point. I accept the answers and agree with the changes in the manuscript. I support the publication of this work in Nature Communications and congratulate the Authors for the method and results that may put BASIC the state of the art.

We would like to thank the Reviewer for her/his valuable comments and support for publication of this work.